# Tumor-Associated Macrophage (TAM)-Related Cytokines, sCD163, CCL2, and CCL4, as Novel Biomarkers for Overall Survival and Time to Treatment in Waldenstrom’s Macroglobulinemia: Emphasis on Asymptomatic WM

**DOI:** 10.3390/cells14040275

**Published:** 2025-02-13

**Authors:** Alexandros Gkiokas, Mavra Papadatou-Gigante, Annita Ioanna Gkioka, Aspasia Koudouna, Thomai M. Tryfou, Alexandros Alexandropoulos, Vassiliki Bartzi, Nikolitsa Kafasi, Marie-Christine Kyrtsonis

**Affiliations:** 1Hematology Section, First Department of Propaedeutic Internal Medicine, Laikon Hospital, Faculty of Medicine, National and Kapodistrian University of Athens, 11527 Athens, Greece; mavra90@yahoo.com (M.P.-G.); anni.iwan.gk@gmail.com (A.I.G.); aspakoud@hotmail.gr (A.K.); thommais@hotmail.com (T.M.T.); al.alexandropoulos@gmail.com (A.A.); vbartzi@yahoo.com (V.B.); mckyrtsonis@gmail.com (M.-C.K.); 2Immunology Department, Laikon Hospital, 11527 Athens, Greece; nkafassi@hotmail.com

**Keywords:** soluble CD163 (sCD163), CCL2 {Monocyte Chemoattractant Protein-1 (MCP-1)}, CCL4 {Macrophage Inflammatory Protein 1 beta (MIP-1b)}, Tumor-associated macrophages (TAMs), Waldenstrom’s Macroglobulinemia (WM), Asymptomatic Waldenstrom’s Macroglobulinemia (AWM), biomarkers, prognosis

## Abstract

Waldenstrom’s Macroglobulinemia (WM) is a heterogeneous disease, and the majority of patients tend to have a long course. Nevertheless, it is imperative to detect patients who have a high risk of progression and who benefit from closer follow-up. Many recent studies have displayed the CD163-positive tumor-associated macrophages (TAMs) contribution in the pathogenesis of various hematological neoplasms and solid tumors. Soluble CD163 (sCD163) can be measured in serum, along with other TAM-chemoattractant cytokines, such as CCL2 and CCL4, and their levels are used to determine macrophage activation. In the current study, we investigated the correlation between sCD163, CCL2, and CCL4, with parameters of WM progression and survival. Out of a total of 204 WM patients, serum sCD163, CCL2, and CCL4 were measured in 75, 64, and 65 patients’ frozen sera at diagnosis, along with 30 healthy individuals (HIs) using an enzyme-linked immunosorbent assay (ELISA). We achieved to demonstrate that shorter Time to Treatment (TTT) was observed in 2 years and 7 years intervals in all patients with a ratio of CD163/CCL4 above median (*p* = 0.003 and *p* = 0.024, respectively) and decreased TTT was observed in all asymptomatic WM (AWM) patients with values of CCL4 above the median (*p* = 0.018). Moreover, significantly decreased overall survival (OS) (*p* = 0.033) was observed in all WM patients with CCL2 values above the median. Our results indicate that sCD163, CCL2, and CCL4 could be utilized as prognostic markers in WM.

## 1. Introduction

Waldenstrom’s Macroglobulinemia (WM) is a rare (1–2% of all hematological malignancies), indolent B-cell lymphoma, with the hallmark of clonal lymphoplasmacytic bone marrow (BM) infiltration (lymphoplasmacytic lymphoma-LPL) and monoclonal IgM secretion [1,2,3]. There are three entities in the spectrum of WM:IgM Monoclonal Gammopathy of Undetermined Significance (IgM-MGUS), asymptomatic WM (AWM), and symptomatic WM [3,4,5,6,7]. In the heterogeneous WM disease spectrum, both IgM-MGUS and AWM are asymptomatic, but AWM is more likely to progress to WM [4]. IgM-MGUS is defined by the presence of monoclonal IgM < 3 g/dL and BM infiltration < 10%. WM is defined by the presence of BM infiltration ≥ 10% with IgM of any concentration, along with the presence of symptoms related to the disease. At the same time, AWM fulfills the diagnostic criteria of WM but requires no treatment initiation [3,4,5,6,7]. Indications for treatment include constitutional symptoms (recurrent fever, night sweats, fatigue, or weight loss), progressive symptomatic lymphadenopathy or splenomegaly, hemoglobin < 10 g/dL or platelets < 100 × 10^9^/L, systemic amyloidosis, signs of hyperviscosity syndrome, and autoimmune manifestations [2,3,4].

The availability of prognostic markers is important in order to discriminate on one hand the subgroup of patients at risk to present with an aggressive disease, and on the other hand detect patients who, although asymptomatic at diagnosis, might need treatment in the near future [1].

Tumor-associated macrophages (TAMs) are macrophages that constitute a vital part of the tumor microenvironment. TAMs are widely present in various tumors, promoting tumor cell growth, dissemination, and drug resistance [8].

TAMs can decompose the extracellular matrix by secreting various enzymes, matrix metalloproteinases (MMPs), and cathepsins, aiding the migration of tumor cells and their extravasation into the circulation [8,9]. Additionally, extravasation is enhanced by increasing the vascular permeability through the secretion of vascular endothelial growth factor (VEGF) and nitric oxide (NO) by TAMs [9,10]. Moreover, they release angiogenetic molecules, including MMP-2, MMP-7, MMP-9, MMP-12, and cyclooxygenase-2 [8,10]. Thus, TAMs not only increase the ability of tumor cells to migrate but also enlarge the actual density of blood vessels [8,9,10].

A plethora of growth factors is secreted by TAMs, such as epithelial growth factor (EGF), platelet-derived growth factor (PDGF), and epithelial growth ligands of the factor receptor (EGFR) family and basic fibroblast growth factor (bFGF), all stimulating tumor cell proliferation. Noteworthy, TAMs are an important cell source for EGF secretion and have an exceptionally important role in the tumor growth of breast and lung cancer, where EGFR ligand is abundant [8].

Moreover, TAMs seem to facilitate the formation of premetastatic niches in distant organs [9,10]. TAM-derived tumor necrosis factor-a (TNF-α), VEGF, and transforming growing factor-β (TGF-β) induce in specific organs the production of S100A8 and serum amyloid A3, both of which can recruit macrophages and tumor cells [10].

With regards to altered immune regulation in tumor microenvironment, TAM-derived IL-6 promotes signal transducer and activator of transcription 3 (STAT3) phosphorylation, leading to decreased apoptosis in tumor cells [10,11]. In addition, they secrete arginase I (ARGI), which metabolizes and depletes L-arginine, an essential amino acid for CD8+ T-cell proliferation. TAMs recruit natural regulatory T (nTreg) cells, further suppressing the antitumor immune activity of CD8+ T-cells [8,9,10,12].

Recruited macrophages most probably account for the majority of TAMs [8,9], while M-MDSCs (monocyte-related myeloid-derived suppressor cells) are another main circulating precursor of TAMs [8]. TAMs receive signals from the microenvironment in which they reside via multiple mechanisms and have mainly the characteristics of M2 macrophages [10]. M2 TAMs express a large number of scavenger receptors, including CD163 [8,11,12]. CD163 is identified as the ‘‘hemoglobin (Hb) scavenger receptor’’ and is responsible for the uptake of Hb released into the plasma. It is expressed almost exclusively on macrophages and monocytes [13,14]. Hence, TAMs are commonly identified with immunochemistry by the expression of CD163 [10]. A soluble variant of CD163 (sCD163), a product of shedding, is present in plasma and other tissue fluids [13] and has been found to be increased in various solid tumors and hematological malignancies, reflecting the increased burden of TAMs overexpressing CD163 [15].

CCL2 Monocyte Chemoattractant Protein-1 (MCP-1) is a member of the C-C chemokine family [16,17], and high levels of CCL2 induce polarization of macrophages to the tumor-promoting phenotype of TAMs [18]. Moreover, CCL4 Macrophage Inflammatory Protein 1 beta (MIP-1b) also belongs to the CC chemokine subfamily [19]. It is secreted by tumor cells together with other chemokines such as CCL2, inducing TAMs infiltration and promoting cancer progression [19,20].

Both CCL2 and CCL4can be secreted by activated leukocytes, lymphocytes, endothelial and cancer cells, and are chemoattractants for natural killer cells and monocytes [16,17,19]. High levels of CCL2 promote polarization of tumor microenvironment macrophages to the phenotype of TAMs and enhance the interaction between them and tumor cells. Additionally, CCL2 may promote macrophages for epithelial-to-mesenchymal transition, making the tumor more prone to metastasis [18].

Recognizing the interplay between sCD163, which acts as a surrogate of CD-163 positive TAMs burden, with CCL2 and CCL4 that enforce TAMs’ tumor infiltration, could reveal new biomarkers. Hence, in our study, we aimed to demonstrate the association between sCD163, CCL2, and CCL4 as prognostic markers in WM, helping to detect high-risk WM patients.

## 2. Materials and Methods

### 2.1. Patient Selection

We retrospectively reviewed 204 WM patients diagnosed in Laikon General Hospital of Athens from 1990 to 2023. As a control group, serum samples from 25 healthy blood donors with a median age of 48 were obtained. The inclusion criteria were a confirmed diagnosis of WM, IgM-MGUS, or LPL according to the WHO 2022 classification, availability of frozen sera samples at the time of diagnosis and before any treatment was administered, and consent of participating in the study. Patients who had received any prior treatment, including corticosteroids before the collection of the serum, or had been diagnosed with solid cancers or autoimmune diseases were excluded from the study since all of these conditions could affect serum CD163 levels.

### 2.2. Cytokine Measurement

Measurements of the cytokines were performed using an enzyme-linked immunosorbent assay (ELISA) (Quantikine, Duo-Set R&D Systems, Minneapolis, MN, USA) according to the manufacturer’s instructions. These essays are solid-phase sandwich ELISA that utilize natural human CD163, CCL2, and CCL4 (catalog numbers DY1607, DY279, and DY271, respectively),with a <0.5% cross-reactivity and no significant interference observed with available related molecules. The sensitivities for CD163, CCL2, and CCL4 are 0.613 ng/mL, 10 pg/mL, and 11 pg/mL, respectively, and the assays’ ranges are 1.6–100 ng/mL, 15.6–1000 pg/mL, and 15.6–1000 pg/mL, respectively.

The method includes plate preparation and sample determination. The microplate was pre-coated with a monoclonal antibody specific for human CD163, CCL2, and CCL4, respectively, and incubated overnight. Plate preparation was performed by diluting the capture antibody to the working concentration in PBS without carrier protein and coating a 96-well microplate with 100 μL per well of the diluted capture antibody. The plate was then sealed and incubated overnight at room temperature. Then, each well was aspirated and washed with wash buffer, repeating the process two times for a total of three washes. Then the plates were blocked by adding 300 μL of reagent diluent to each well and incubated at room temperature for a minimum of 1 h. After the plates were again aspirated and washed, 100 μL of sample or standards in reagent diluent were added per well and incubated for 2 h at room temperature. The aspiration/wash process was repeated, and 100 μL of the detection antibody, diluted in reagent diluent, was added to each well and incubated for 2 h at room temperature. After an aspiration/wash process, 100 μL of the working dilution of Streptavidin-HRP was added to each well and incubated for 20 min at room temperature. The aspiration/wash process was repeated, and 100 μL of substrate solution was added to each well andincubated for 20 min at room temperature, avoiding the placement of the plate in direct light. Finally, 50 μL of stop solution was added to each well, and the plates were gently tapped to ensure thorough mixing. The optical density of each well was immediately determined using a microplate reader set to 450 nm. All measurements were performed in duplicate. Unfortunately, all WM patients showed levels of sCD163, surpassing the highest standard value. Hence, it was required to perform dilution of the sample with a diluent and repeat the assay, utilizing a dilution ratio of 1:4. All cytokine levels were expressed in picogram/mL.

### 2.3. Statistical Analysis

For the purposes of statistical analysis, the SPSS v.28 software was utilized. Kaplan–Meier curves showed overall survival, and the differences in outcomes between the studied subgroups were estimated using the log-rank test. Statistical significance was set at *p* < 0.05. To compare plasma levels of cytokines from controls versus patients, statistical analysis was performed using the Mann–Whitney test, and *p* values < 0.05 were considered significant. Considering the number of the studied patients, we utilized the median values as the optimal cut-off for statistical analysis since they can dichotomize a continuous variable and guarantee an equal sample size for both groups. Additionally, since there are no known normal ranges for these cytokines, we attempted to find a cut-off that would provide statistical significance. Spearman’s test was used to analyze the correlation between various cytokines.

### 2.4. Study Approval

This study was approved by the institutional review board of Laikon General Hospital of Athens with protocol number 2782, and written informed consent was obtained from all participants in research for analysis of medical history and sera collection. Patient confidentiality and data anonymization were maintained throughout the study.

## 3. Results

Two hundred and four patients were included in the study, of whom 82 had WM, 88 had AWM, 14 had IgM-MGUS, and 20 had LPL. 56% were males, and 44% were females, with a mean age of 66.5 years (range, 33–92 years). As AWM patients were defined as those fulfilling the diagnostic criteria for WM but not requiring treatment at diagnosis, the term also includes patients receiving treatment 6 months or more after the initial diagnosis. Time to Treatment (TTT) was defined as the interval between the initial diagnosis and treatment initiation.

The patients’ characteristics and their laboratory tests at diagnosis are shown in Table 1. The median levels at the time of diagnosis for platelets, hemoglobin, white blood cells, lymphocytes, monocytes, IgM, LDH, b2-microglobulin and ESR were 215 K/μL, 11.3 gr/dL, 7.14 K/μL, 6.7 K/μL, 2 K/μL, 0.49 K/μL, 2088 mg/dL, 290 IU/L, 3.29 mg/dL and 86.5 mm/hr, respectively.

In our study, median values of the cytokines’ measurements were chosen as cut-offs for statistical analysis. sCD163 was measured in 75 patients and the median value was 28,163 pg/mL (range: 16,696 to 97,286 pg/mL) in WM, and 27,368 pg/mL (range: 25,410 to 51,319 pg/mL) in patients with LPL; it was statistically significantly higher than in patients with IgM-MGUS (median:26,821 pg/mL (range: 14,281 to 97,280 pg/mL) and healthy individuals (HI) (median: 26,826 pg/mL; range: 11,831–97,286 pg/mL) (*p* < 0.001). Serum CCL2 was measured in 64 patients and the median value was 347.5 pg/mL (range: 291 to 1829 pg/mL) in HI and 497.45 pg/mL (range: 6.64 to 1713.11 pg/mL) in patients; serum CCL4 was measured in 65 patients the median value was 202 pg/mL (range: 185.53 to 578.61 pg/mL) in HI and 278.61 pg/mL (range: 0–2462 pg/mL) in patients (Table 2).

Both the 2-year and 7-year TTT were shorter in all patients with a ratio of CD163/CCL4 above median (*p* = 0.003 and *p* = 0.024, respectively) (Figure 1 and Figure 2). Additionally, significantly decreased TTT was observed in all AWM patients with values of CCL4 above the median (*p* = 0.018) (Figure 3). Patients with values of sCD163 and CCL2 above the median did not show a shorter TTT in 2-or 7-year intervals (Appendix A).

A statistically significant decreased Overall Survival (OS) (*p* = 0.033) (Figure 4) was observed in all WM patients with CCL2 values above median. In AWM patients, CCL2 above median had a strong tendency for decreased OS (*p* = 0.08) (Appendix A). sCD163 and CCL4 did not show an OS difference when the median value was used as the cut-off point to discriminate the patients’ groups (Appendix A). It is worth noticing that the absolute number of monocytes count and the ratio of monocytes-to-lymphocytes were not associated with OS or TTT.

Correlation bivariate analysis amongst sCD163, CCL2, and CCL4 demonstrated a positive correlation by Spearman between CCL2 and CCL4 (Spearman correlation coefficient: 0.433, *p* < 0.001) and CCL2 with the ratio sCD163/CCL4 (Spearman correlation coefficient: 0.461, *p* < 0.001). Similarly, correlation analysis between sCD163, CCL2 and CCL4, and other cytokines known to have a prognostic impact in WM pathogenesis showed the following positive correlations; sCD163 with TGFb (0.343, *p* = 0.037), sCD163 with serum Syndecan (CD138) (0.326, *p* = 0.034), CCL4 with VEGF (0.394, *p* = 0.023), and CD163/CCL4 with VEGF (0.453, *p* = 0.011). Moreover, we examined the correlation of sCD163, CCL2 and CCL4 with clinical factors of WM and the following correlations were revealed; bone marrow infiltration was correlated with both sCD163 (0.249, *p* = 0.04) and CCL2 (0.272, *p* = 0.043), and occurrence of splenomegaly was associated with CCL4 (0.311, *p* = 0.023).

Mann-Whitney test revealed the following results between the cytokines tested and known prognostic factors for WM; CCL2 and bone marrow infiltration greater than 60% (*p* = 0.05), CCL4 and PLTs below 100.000/μL (*p* = 0.001), CCL4 and the ratio of monocyte/lymphocyte above the median value (*p* = 0.008), CD163 and IgM above the median value (*p* = 0.027).

## 4. Discussion

In our study, two hundred and four patients were included, of whom 82 were diagnosed with WM, 88 with AWM, 14 with IgM-MGUS, and 20 with LPL.

WM mortality is mainly attributed to disease-related symptoms and complications; thus, IgM-MGUS and AWM have an OS similar, or slightly decreased, compared to the general population and can remain stable for years [6]. Conversely, symptomatic WM has a median overall survival that ranges from 5 years to more than 10 years [2,6,21,22]. OS was not improved drastically after the introduction of the rituximab-based regimen in WM, an observation mainly attributed to the low rate of patients achieving Complete Response (CR) [23,24]. Therefore, over the past few years, many studies have focused on associating patients’ characteristics and biomarkers with their clinical outcome, attempting to detect patients who present with adverse characteristics [1,21,25]. However, the rarity and the heterogeneity of WM, along with the long follow-up needed to extract safe conclusions, make the development of new biomarkers challenging [6,21].

TAMs and their contribution to the tumor microenvironment have been studied for years. They secrete numerous cytokines that stimulate tumor cell proliferation and survival, with the enhancement of invasion/metastasis and the creation of premetastatic niches being the most thoroughly described mechanism by which TAMs promote solid tumor progression [8,10].

As more light has been shed on the relationship between TAMs and neoplasms, their level of infiltration has begun to be used as a potential biomarker for diagnosis and prognosis. Recently, various studies have established that TAMs and sCD163 have a crucial role in the pathogenesis of many hematopoietic malignancies such as Multiple myeloma (MM) [15,26,27], Hodgkin lymphoma (HL) [28,29], Chronic Lymphocytic leukemia (CLL) [30], and Diffuse B-cell large lymphoma (DLBCL) [31,32]. In all the aforementioned studies, increased sCD163 evaluated with ELISA and CD163+ TAMs tumor infiltration evaluated with immunohistochemistry were indicative of unfavorable prognosis, poorer outcome, rapid progression, and increased likelihood of treatment resistance or recurrence.

The vast majority of patients in our study with the diagnosis of WM/AWM had elevated levels of serum sCD163, regardless of their clinical and laboratory findings. Consequently, increased levels of sCD163 are a common finding in patients with WM, implicating that TAMs play a crucial role in disease pathogenesis. Notably, IgM-MGUS patients and HI had similar values and median values of sCD163. Since IgM-MGUS is a naive condition with a very low annual incidence of progression to WM (6), we assume that the infiltration by TAMs should be minimal. Hence, the sCD163 levels are not elevated. This observation further supports the hypothesis of TAMs being involved in WM disease progression. The fact that the levels of the sCD163 in the group of HI with a median age of 48 did not differ from the levels measured in the group of IgM-MGUS patients who had a considerably higher median age indirectly shows that there is no relationship between advanced age and elevations of sCD163.

TAMs receive signals for their polarization from the microenvironment in which they reside and are not observed in the steady state but in pathologic conditions, such as neoplasms [10,11]. CCL2 and CCL4 are cytokines that belong to the C-C chemokine family characterized by adjacent cysteine residue [16,17,19]. Tumor-cell-derived CCL4 and CCL2 can enhance TAM infiltration, leading to cancer progression. The expression levels of both CCL4 and CCL2 were elevated in colon-cancer and lung adenocarcinoma tissues and were associated with shorter OS [19,20,33]. Notably, the Inhibition of CCL2 and CCL4 expression has been shown to suppress macrophage infiltration in solid tumors [19,33].

To our knowledge, the data on the hematological neoplasms are scarce, and no other data on WM, evaluating CCL2 and CCL4 as chemoattractant cytokines for TAM recruitment, along with CD163. It is vital to recognize not only the sCD163 as a possible biomarker in WM but also other cytokines known to increase BM infiltration of TAMs, such as CCL2 and CCL4, as they could be used as potential therapeutic targets [19].

In our study, we showed that serum CCL4 and its ratio with CD163 (CD163/CCL4) were able to predict TTT in WM and AWM, as patients with values above the median had significantly shortened TTT. Moreover, AWM with CCL4 values above the median demonstrated a more rapid evolution to symptomatic WM, as was depicted by their shortened TTT. However, patients with values of sCD163 and CCL2 above the median did not show a shorter TTT in 2- or 7-year intervals (Appendix A). The rationale of utilizing the CD163/CCL4 ratio is based on TAMs pathophysiology; sCD163 seems to be a reliable indicator of TAMs’ burden [15], and CCL4 constitutes a strong TAMs’ chemoattractant [19,20]. Further investigation is needed to verify our finding. However, it would imply that serum sCD163 levels, CCL4 levels, and their ratio can eventually discriminate patient with AWM who will evolve to symptomatic WM, guiding decisions regarding the intervals of their follow-up. Additionally, AWM and WM patients with elevated serum CCL2 seem to have a diminished OS, revealing a probable new OS-biomarker in a long-standing disease. sCD163 and CCL4 failed to show an OS difference when the median value was used as the cut-off point to discriminate the patients’ groups (Appendix A).

Tumor microenvironment has been well studied in WM, and various cytokines have been deemed to orchestrate the pathogenetic mechanisms, such as IL-1, sIL-6, Syndecan (CD138), and TGF-b [34,35,36,37]. Our analysis revealed a strongly positive correlation not only amongst CCL2, CCL4, and CD163 but also with the other cytokines involved in WM pathogenesis, showing that TAMs and the related cytokines should have a crucial role in the disease course.

Biomarkers display an important tool in medicine, facilitating diagnosis and prognosis, and are useful in numerous ways, including measuring disease progression, detecting high-risk patients for treatment failure, and establishing susceptibility to recurrence. WM is a rare and indolent disease;hence, long-term follow-up is needed to exclude safe results regarding new biomarkers. We achieved to demonstrate in a large group of WM patients with a very long follow-up that sCD163, CCL2, and CCL4, readily measured with ELISA, may be used as biomarkers to predict TTT and OS, reinforcing the hypothesis that TAMs play an important role in disease pathogenesis. These cytokines could be used in everyday clinical practice and be integrated in established prognostic models for WM and AWM to distinguish the patients that exhibit high-risk features for evolution to symptomatic disease that requires treatment. AWM patients with higher levels of CCL4 and CD163/CCL4 could benefit from a closer follow-up, as according to our study, they tend to have a significantly shorter TTT compared to patients with lower levels.

## 5. Conclusions

In our study, we demonstrated that sCD163, along with CCL2 and CCL4, could be utilized as novel biomarkers in WM, predicting TTT, especially in AWM and OS. Since the levels of sCD163 act as surrogates of the TAMs’ tumor burden, and CCL2 along with CCL4 are well-studied chemoattractant cytokines for TAMs’ infiltration, we hypothesize that TAMs play an important role in WM pathogenesis. The roles of CD163, CCL2, and CCL4 in WM pathogenesis seem to be interwoven and mutually affecting each other, as their values and ratios seem to be prognostically important. Hence, ensuing studies are required to establish our findings and fully clarify their interactions on gene and molecular levels.

## Figures and Tables

**Figure 1 cells-14-00275-f001:**
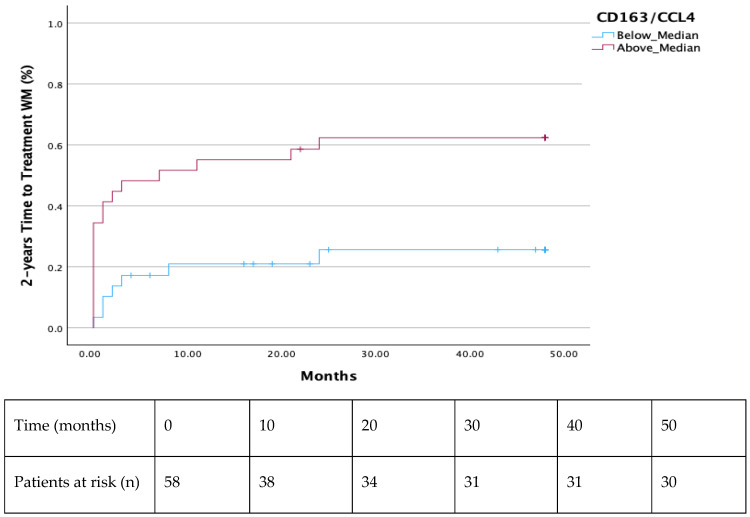
Shorter 2-year TTT in all patients with a ratio of CD163/CCL4 above the median (*p* = 0.003).

**Figure 2 cells-14-00275-f002:**
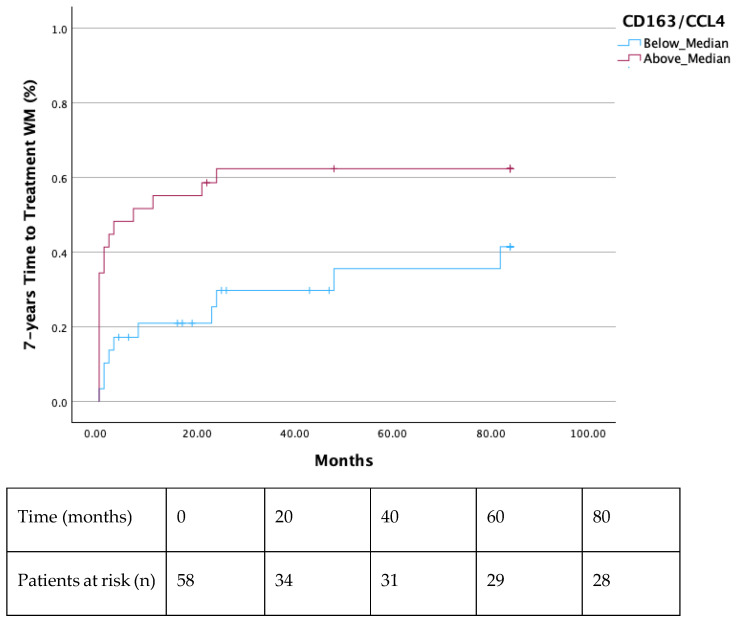
Shorter 7-year TTT in patients with a ratio of CD163/CCL4 above the median (*p* = 0.024).

**Figure 3 cells-14-00275-f003:**
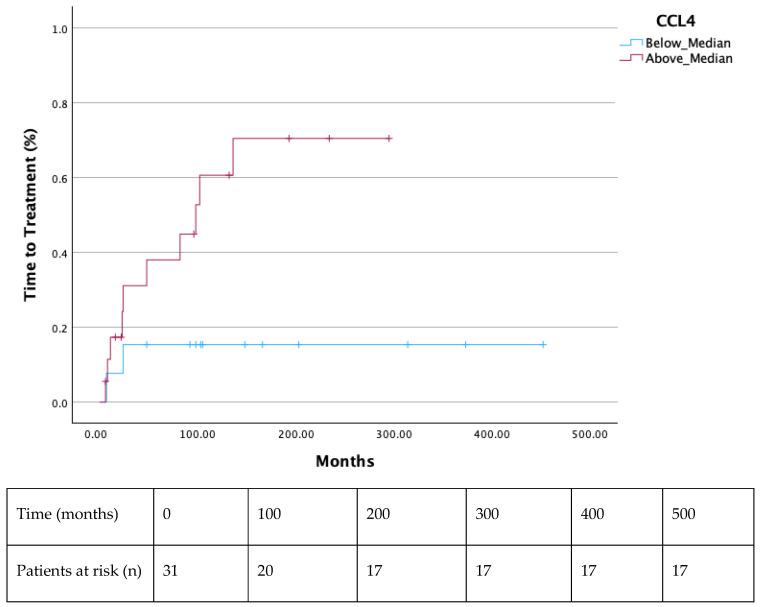
Shorter TTT in AWM patients with CCL4 above the median (*p* = 0.018).

**Figure 4 cells-14-00275-f004:**
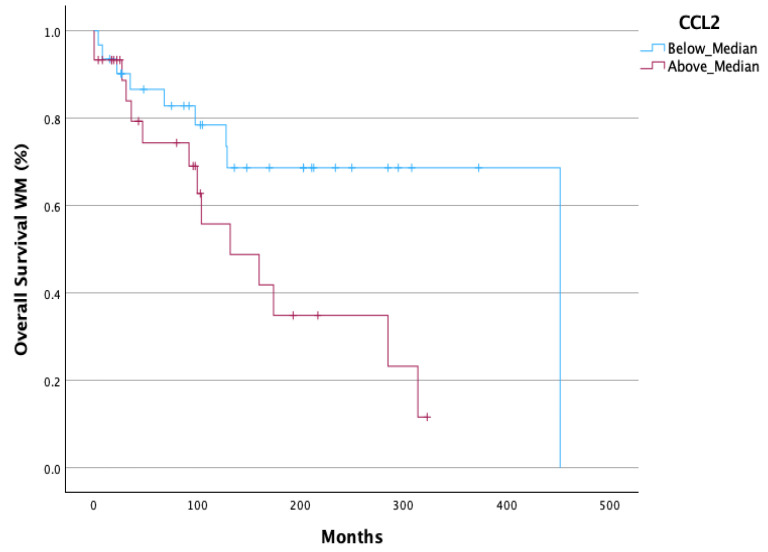
Decreased OS in all WM patients with a value of CCL2above the median (*p* = 0.033).

**Table 1 cells-14-00275-t001:** Clinical and laboratory parameters of the total study group and the subgroup of patients with available sera for cytokine measurement.

Clinical and Laboratory Parameters	Patients	Patients with Available Serum at Diagnosis
Total	204	75
Median age	66.5 years (range, 33–92 years)	64 (33–92)
Gender		
Female	44%	48%
Male	56%	52%
Diagnosis		
WM	40%	37%
AWM	43%	44%
IgM-MGUS	7%	11%
LPL	10%	8%
Free-light chain type	n = 158	n = 67
IgM-kappa	120	76
IgM-lamda	34	19.5
Biclonal	4	4.5
Median BM infiltration	n = 177	n = 68
	40% (range: 0–40)	37.5% (range: 5–90)
Median IgM (mg/dL)	n = 187	n = 72
	2088 (range: 38–12,300)	1365 (range: 38.2–11,040)
Presence of Lymphadenopathy	n = 179	n = 65
	21.20%	13.80%
Presence of Organomegaly	n = 171	n = 62
	16.40%	11.30%
Median β2-microglobulin (mg/dL)	n = 117	n = 63
	3.29 (range: 0.59–16.7)	3 (range: 0.59–16.3)
Median Erythrocyte sedimentation rate (ESR) (mm/h)	n = 116	n = 52
	86.5 (range: 5–150)	68 (range: 10–150)
Median Platelet count (K/μL)	n = 194	n = 73
	215 (range: 4–489)	214 (range: 16–490)
Median Albumin (g/dL)	n = 195	n = 74
	4 (range: 1.5–4.5)	4.2 (range: 2–4.5)
Median Total protein (g/dL)	n = 151	n = 67
	8.1 (range: 4.86–12.3)	7.9 (range: 5.7–11.9)
Median Hemoglobin (g/dL)	n = 194	n = 73
	11.3 (range: 4.9–15.4)	11.7 (range: 5.7–15.2)
Median White Blood Leucocytes absolute count (K/μL)	n = 193	n = 73
	6.7 (range: 2.2–52.7)	6.7 (range: 2.02–23)
Median Lymphocyte absolute count (K/μL)	n = 193	n = 73
	2.0 (range: 0.66–40.6)	2.0 (range: 0.22–19.3)
Median Monocyte absolute count (K/μL)	n = 117	n = 53
	0.49 (range: 0.05–5.2)	0.47 (range: 0.05–2.17)
Median LDH (IU/L)	n = 174	n = 71
	290.5 (range: 3–1150)	289 (range: 73–1150)

**Table 2 cells-14-00275-t002:** Results of cytokine measurements in all WM patients.

	Number of Patients	Median Value	Range
sCD163	n = 75	28,163 pg/mL	16,696–97,286 pg/mL
CCL2	n = 64	497.45 pg/mL	6.64–1713.11 pg/mL
CCL4	n = 65	278.61 pg/mL	0–2462 pg/mL

## Data Availability

Data are contained within the article.

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
