# Peer review of "Tumor-Associated Macrophage (TAM)-Related Cytokines, sCD163, CCL2, and CCL4, as Novel Biomarkers for Overall Survival and Time to Treatment in Waldenstrom’s Macroglobulinemia: Emphasis on Asymptomatic WM"

_cells, 2025, doi:10.3390/cells14040275_

Round 1
Reviewer 1 Report
Comments and Suggestions for Authors
Overall: This paper investigated the correlation of soluble CD162, CLL2 and CLL4, as TAMs-related cytokines, to WM progression and survival. By analyzing the serum levels of these cytokines as novel biomarkers in WM patients, authors were trying to establish predicting biomarkers that are related to treatment and over survival of WM patients.
Comments:
1. Table 2 showed that the number of examined patients for these 3 cytokines measurements are different from each other. Please explain why and what parameters were used to select patient cohort for each cytokine. Paper addressed that there was a total of 204 WM patients, would think that these cytokines would be measured for all patient samples. Please explain how patient cohort was selected for each target, instead of using the whole sample group for all three targets.
2. Figure 1 and 2 showed that shorter 2- and 7-year TTT in patients with a ratio of CD163/CCL4 above the median. How about the level of each cytokine? Do they also show the same/similar results to the TTT? Authors studied serum levels of all three cytokines, seems odd without showing the correlation between serum level of each cytokine and TTT. Please show figures and interpret results.
Why choose the ratio of CD163/CCL4? What is the significance of this ratio? Please explain. How about the ratio of CD163/CCL2?
3. Figure 3 showed shorter TTT in AWM patients with high CLL4 level. Does this shorter TTT also showed in all WM patients? Please add figures and explain.
What about CD163 and CLL2 levels in AWM and all WM patients? And why this time not using the ratio of CD163/CCL4 or CD163/CCL2? Please explain.
4. Figure 4 showed decreased OS in all WM patients with CLL2 level above the median. How about the level of CD163 and CCL4? Please add figures and explain results. And again, why not using the ratio for this? Please explain.
Please explain why switching and showing different parameters when demonstrating the correlation of these cytokines to TTT and OS, and in different patient cohorts. Seems missing data to fully address the conclusions.
Author Response
Comment 1: Table 2 showed that the number of examined patients for these 3 cytokines measurements are different from each other. Please explain why and what parameters were used to select patient cohort for each cytokine. Paper addressed that there was a total of 204 WM patients, would think that these cytokines would be measured for all patient samples. Please explain how patient cohort was selected for each target, instead of using the whole sample group for all three targets.
Response 1: The number of the patients that were examined for each cytokine was clearly dictated by the availability of the sera. Our database and biobank were created in 1990s and patients are constantly added. However, since many experiments have been undertaken since then, many of the sera at the time of diagnosis have been exhausted.
We clearly state in lines 169-178 the number of the patients that we tested for each cytokine. The number 204 is the total number of our cohort.
Comment 2: Figure 1 and 2 showed that shorter 2- and 7-year TTT in patients with a ratio of CD163/CCL4 above the median. How about the level of each cytokine? Do they also show the same/similar results to the TTT? Authors studied serum levels of all three cytokines, seems odd without showing the correlation between serum level of each cytokine and TTT. Please show figures and interpret results.Why choose the ratio of CD163/CCL4? What is the significance of this ratio? Please explain. How about the ratio of CD163/CCL2?
Response 2: We performed statistical analysis for each cytokine and OS and TTT. Unfortunately, we present only the results that show statistical significance (p<0.005). Would you like to add as a supplementary material the statistical analysis performed using each cytokine and the results regarding TTT, even though no statistical significance is shown? The rationale of the ratio CD163/CCL4 is based on the pathophysiology of TAMs, and we were really excited by its used and the fact that it provided results that were statistically significant. Thank you for pointing that out. CD163 is expressed on TAMs and constitutes a very specific marker of expression. On the other hand CCL4 acts as chemoattractant for TAMs and promotes their expansion. Lines 88 and 102 in the introduction provide the relevant information regarding the pathophysiology.
Comment 3: Figure 3 showed shorter TTT in AWM patients with high CLL4 level. Does this shorter TTT also showed in all WM patients? Please add figures and explain. What about CD163 and CLL2 levels in AWM and all WM patients? And why this time not using the ratio of CD163/CCL4 or CD163/CCL2? Please explain.
Response 3: Again we would like to highlight that we opted to demonstrate only the results that were statistically significant for each patients' group, otherwise the information would be overwhelming. However, we would be glad to provide as supplementary information all the other data if you consider that crucial.
Comment 4: Figure 4 showed decreased OS in all WM patients with CLL2 level above the median. How about the level of CD163 and CCL4? Please add figures and explain results. And again, why not using the ratio for this? Please explain.Please explain why switching and showing different parameters when demonstrating the correlation of these cytokines to TTT and OS, and in different patient cohorts. Seems missing data to fully address the conclusions.
Response 4: Thank you for pointing out the potential relevance of all the cytokines. We would like to mention, as it is stated in our article, that it is the first time that a team attempts to correlate TAMs-associated cytokines with WM pathogenesis and further study is needed. However, we are confident to say that in a rather big patients' group for the specific disease (considering its rarity), we achieved to show that these cytokines have an impact on OS and TTT.
We used 2 different cohorts because WM and AWM constitute two different entities. AWM is a condition that requires no treatment and patients can remain stable for years. In the recent years, many prognostic scores have been develop to predict the course of AWM patients and our team has created one as well.
Reviewer 2 Report
Comments and Suggestions for Authors
The authors investigated the prognostic role of TAM-related sCD163 and CCL2/4 in blood samples from patients with WM (and their subgroups).
They succeeded in demonstrating that sCD163, along with CCL2 and CCL4, could be utilized as novel biomarkers in WM to predict TTT, especially in AWM and OS. They think that TAMs play a big part in the development of WM because sCD163 levels show how many TAMs are in the tumor microenvironment along with CCL2 and CCL4, which are well-studied chemoattractant cytokines that TAMs use to get inside.
It is an intriguing study with a relatively high number of involved patients. The ethical issues are correct.
However, I have some aspects that need revisions.
My questions are the following:
TAMs originate from cells of innate immunity, whereas WM is a plasma cell dyscrasia.
What is the logical relationship between macrophages and plasma cells that led to the study of TAM-related markers?
Have TAMs been found in bone marrow biopsy specimens within WM groups? Did these TAMs express CD163, CCL2, or CCL4?
What symptoms did WM patients exhibit? How did they distinguish WM patients from the autoinflammatory disease with IgM monoclonality, Schnitzler syndrome? How can they be sure that some of the patients they tested were not Schnitzler's? In the latter case, the presence of activated macrophages seems more logical because of the autoinflammatory nature of the disease.
There are several typos or non-syntactically incorrect passages in the abstract and the text that I consider corrections.
Author Response
Comment 1: TAMs originate from cells of innate immunity, whereas WM is a plasma cell dyscrasia.What is the logical relationship between macrophages and plasma cells that led to the study of TAM-related markers?
Response 1: Thank you for the interesting comment. As it is stated in the Discussion in line 228 "TAMs and their contribution in the tumor microenvironment have been studied for years. They secrete numerous cytokines that stimulate tumor cell proliferation and survival, with the enhancement of invasion/metastasis and the creation of premetastatic niches being the most thoroughly described mechanism by which TAMs promote solid tumor progression.As more light has been shed on the relationship between TAMs and neoplasms, their level of infiltration has begun to be used as potential biomarker for diagnosis and prognosis. In recent years, several studies have demonstrated the roles of TAMs and sCD163 in the pathogenesis of hematopoietic malignancies including Multiple myeloma (MM), Hodgkin lymphoma (HL), Chronic Lymphocytic leukemia (CLL), and Diffuse B-cell large lymphoma (DLBCL) [31,32]. In all the aforementioned studies, increased sCD163 evaluated with ELISA and CD163+ TAMs tumor infiltration evaluated with immunohistochemistry, were indicative of unfavorable prognosis, poorer outcome, rapid progression, and increased likelihood of treatment resistance or recurrence.
Comment 2: Have TAMs been found in bone marrow biopsy specimens within WM groups? Did these TAMs express CD163, CCL2, or CCL4?
Response 2: Unfortunately, since our study is retrospective we did not have the chance to study the bone marrow biopsies of our patients regarding the CD163 expression. However, we are confident from the bibliography and the relevant studies that we mentioned, that the level of sCD163 correlated with CD163+ macrophages of the BM since CD163 is a rather specific marker for TAMs.
Comment 3: What symptoms did WM patients exhibit? How did they distinguish WM patients from the autoinflammatory disease with IgM monoclonality, Schnitzler syndrome? How can they be sure that some of the patients they tested were not Schnitzler's? In the latter case, the presence of activated macrophages seems more logical because of the autoinflammatory nature of the disease.
Response 3: Thank you for your comment. Patients in our cohort fulfilled the diagnostic criteria of WM, and not Schnitzler's syndrome, based on WHO 2022. Even though there is an overlap regarding the clinical presentation, we strictly follow the diagnostic process/criteria used worldwide. As we state in our text, in line 118 "Patients who had received any prior treatment including corticosteroids before the collection of the serum or had been diagnosed with other malignancies or autoimmune diseases that could influence serum sCD163 levels, were excluded from the study.". Your observation is really interesting, since we can observe in our study that WM, as other hematologic malignancies correlate with TAMs' overexpression. It seem to the pro-tumor activity of TAMs, rather than the inflammatory role, that promotes the tumor growth, both for solid and hematologic malignancies.
Round 2
Reviewer 1 Report
Comments and Suggestions for Authors
Thanks for authors' quick response.
For figures, I understand that authors presented only the results that show statistical significance. That's also my first guess when did not see figures for correlations of each cytokine to TTT/OS. Also agreed with authors that furthermore study is needed to fully address the correlation of TAMs’-associated cytokines and WM pathogenesis.
Authors should discuss these results for each cytokine and OS and TTT in Discussion section, to address that even though there is no significant direct correlation between each examined cytokine level and TTT/OS, but authors did find that the ratio of CD163/CCL4 is strongly correlated. This kind of discussion would highlight the significance of using the ratio of CD163/CCL4 as biomarker in WM study and provide more guildlines for future related studies.
It would be nice if authors can add these results as supplementary materials. Only showing statistical significance results without discussing the rest of the results doesn't feel like reading a full story of the study. It seems odd that authors clearly have performed these studies, but without showing or discussing it.
Strongly recommend authors to add a discussion part to discuss these results, along with addressing the rational of using the ratio of CD163/CCL4. It likes "Although we did not observe this..., we did find that...". It would help to address the conclusions and complete a full story, provide more comprehensive understanding of the results and point out directions for future studies.
A thought thrown out:
Comparing results showed between figure 1/2 and 3, it looks like CCL4 is more correlated to WM disease progression. As authors addressed, AWM is a WM condition requires no treatment, but high levels of CCL4 leads to shorter TTT in AWM patients (fig 3). Then a high ratio of CD163/CCL4 leads to shorter TTT in all WM patients (figure1 and 2). Seems like CCL4 is playing some roles in accelerating WM disease progression. It would be clearer if authors can compare CCL4 level changing in AWM patients initially then progressed to WM. Just some thoughts for “Could be” future directions.
Author Response
Comment: Authors should discuss these results for each cytokine and OS and TTT in Discussion section, to address that even though there is no significant direct correlation between each examined cytokine level and TTT/OS, but authors did find that the ratio of CD163/CCL4 is strongly correlated. This kind of discussion would highlight the significance of using the ratio of CD163/CCL4 as biomarker in WM study and provide more guildlines for future related studies. It would be nice if authors can add these results as supplementary materials. Only showing statistical significance results without discussing the rest of the results doesn't feel like reading a full story of the study. It seems odd that authors clearly have performed these studies, but without showing or discussing it.Strongly recommend authors to add a discussion part to discuss these results, along with addressing the rational of using the ratio of CD163/CCL4. It likes "Although we did not observe this..., we did find that...". It would help to address the conclusions and complete a full story, provide more comprehensive understanding of the results and point out directions for future studies.
Response: Thank you for the very useful comment. We totally agree with you that providing also the "negative"/not-statistically important results enhances the discussion section and provides a better view of the extensive study we have performed on WM patients. Hence, we mention all the results in lines 273 and 282, as you suggested. Furthermore, we provide a supplementary material file with the statistical analysis. Thank you once again for the corrections.
Reviewer 2 Report
Comments and Suggestions for Authors
I accept the answers of the authors.
Author Response
Comment: I accept the answers of the authors.
Response: Thank you for your useful comments.